# Systematic Comparison of Hospital-Wide Standard and Model-Based Therapeutic Drug Monitoring of Vancomycin in Adults

**DOI:** 10.3390/pharmaceutics14071459

**Published:** 2022-07-13

**Authors:** Heleen Gastmans, Erwin Dreesen, Sebastian G. Wicha, Nada Dia, Ellen Spreuwers, Annabel Dompas, Karel Allegaert, Stefanie Desmet, Katrien Lagrou, Willy E. Peetermans, Yves Debaveye, Isabel Spriet, Matthias Gijsen

**Affiliations:** 1Pharmacy Department, UZ Leuven, 3000 Leuven, Belgium; heleen.gastmans@uzleuven.be (H.G.); ellen.spreuwers@uzleuven.be (E.S.); isabel.spriet@uzleuven.be (I.S.); 2Clinical Pharmacology and Pharmacotherapy, Department of Pharmaceutical and Pharmacological Sciences, KU Leuven, 3000 Leuven, Belgium; erwin.dreesen@kuleuven.be (E.D.); nada.dia@kuleuven.be (N.D.); karel.allegaert@kuleuven.be (K.A.); 3Department of Clinical Pharmacy, Institute of Pharmacy, University of Hamburg, 20146 Hamburg, Germany; sebastian.wicha@uni-hamburg.de; 4Department of Information Technology, University Hospitals Leuven, 3000 Leuven, Belgium; annabel.dompas@uzleuven.be; 5Department of Development and Regeneration, KU Leuven, 3000 Leuven, Belgium; 6Department of Hospital Pharmacy, Erasmus MC University Medical Center, 3015 GD Rotterdam, The Netherlands; 7Laboratory of Clinical Bacteriology and Mycology, Department of Microbiology, Immunology and Transplantation, KU Leuven, 3000 Leuven, Belgium; stefanie.desmet@uzleuven.be (S.D.); katrien.lagrou@uzleuven.be (K.L.); 8Department of Laboratory Medicine, UZ Leuven, 3000 Leuven, Belgium; 9Laboratory of Clinical Infectious and Inflammatory Disease, Department of Microbiology, Immunology and Transplantation, KU Leuven, 3000 Leuven, Belgium; willy.peetermans@uzleuven.be; 10Department of General Internal Medicine, UZ Leuven, 3000 Leuven, Belgium; 11Laboratory for Intensive Care Medicine, Department of Cellular and Molecular Medicine, KU Leuven, 3000 Leuven, Belgium; yves.debaveye@uzleuven.be

**Keywords:** vancomycin, therapeutic drug monitoring, population pharmacokinetics, precision dosing, predictive performance, model averaging, model selection, Bayesian forecasting

## Abstract

We aimed to evaluate the predictive performance and predicted doses of a single-model approach or several multi-model approaches compared with the standard therapeutic drug monitoring (TDM)-based vancomycin dosing. We performed a hospital-wide monocentric retrospective study in adult patients treated with either intermittent or continuous vancomycin infusions. Each patient provided two randomly selected pairs of two consecutive vancomycin concentrations. A web-based precision dosing software, TDMx, was used to evaluate the model-based approaches. In total, 154 patients contributed 308 pairs. With standard TDM-based dosing, only 48.1% (148/308) of all of the second concentrations were within the therapeutic range. Across the model-based approaches we investigated, the mean relative bias and relative root mean square error varied from −5.36% to 3.18% and from 24.8% to 28.1%, respectively. The model averaging approach according to the squared prediction errors showed an acceptable bias and was the most precise. According to this approach, the median (interquartile range) differences between the model-predicted and prescribed doses, expressed as mg every 12 h, were 113 [−69; 427] mg, −70 [−208; 120], mg and 40 [−84; 197] mg in the case of subtherapeutic, supratherapeutic, and therapeutic exposure at the second concentration, respectively. These dose differences, along with poor target attainment, suggest a large window of opportunity for the model-based TDM compared with the standard TDM-based vancomycin dosing. Implementation studies of model-based TDM in routine care are warranted.

## 1. Introduction

Vancomycin is a glycopeptide antibiotic, administered intravenously to treat severe infections due to Gram-positive bacteria. Vancomycin is widely used across different patient populations [1]. Considering its relatively narrow therapeutic range, plasma concentrations are usually monitored during therapy, and therapeutic drug monitoring (TDM) is ubiquitously applied to guide vancomycin dosing [2,3].

Until recently, most institutions used concentrations obtained during intermittent (trough concentration, C_trough_) or continuous infusion to guide vancomycin dosing as was recommended in the first consensus guideline for the TDM of vancomycin in 2009 [3]. In the 2020 update of this guideline, the ratio of the area under the concentration–time curve over 24 h to the minimum inhibitory concentration (AUC/MIC) was recommended over the use of vancomycin concentrations to guide the treatment of serious infections caused by methicillin-resistant *Staphylococcus aureus* [2]. Despite this recommendation, the pharmacokinetic/pharmacodynamic (PK/PD) index of interest that should be used to guide vancomycin dosing is still under debate [4,5,6,7]. Some argue that the evidence for using AUC/MIC-based TDM over the monitoring of vancomycin concentrations is inconclusive and that it is too early to justify the increased resource use associated with AUC/MIC monitoring [4,6,8]. As a result, there has not been a universal application of AUC/MIC-based TDM of vancomycin to date, and single vancomycin concentrations are still regularly used to guide dosing.

Irrespective of the PK/PD index, evidence reveals that many patients still experience vancomycin exposure outside of the therapeutic range with standard TDM-based strategies, associated with either reduced efficacy or increased toxicity [9,10]. Therefore, Bayesian forecasting has been proposed to provide standardized and accurate dose adaptations following the TDM of vancomycin [2,10]. Bayesian forecasting relies on the combination of previously developed pharmacokinetic population (pop) PK models and patient information, such as plasma concentrations, previous dosing information, and patient characteristics. The individually generated PK estimates then allow the prediction of future exposure according to specific dosing adaptations. This whole process is commonly referred to as model-informed precision dosing (MIPD) [11].

Currently, several dosing software modules offer vancomycin MIPD [11]. However, the accuracy of model-based dosing recommendations depends on the PopPK model used [12]. Several studies illustrated that it is particularly challenging to select one appropriate PopPK model of vancomycin for use in real-world clinical practice [13,14]. Recently, Uster et al. suggested two multi-model approaches—a model selection algorithm (MSA) and a model averaging algorithm (MAA)—that might address this important challenge and stimulate the integration of model-based vancomycin dosing in clinical practice [15].

Therefore, in this study, we aimed to evaluate the predictive performance and the predicted doses of a single-model approach and several multi-model approaches based on seven vancomycin models for the Bayesian forecasting of vancomycin dosing compared with the standard TDM-based dosing in a hospital-wide setting for both intermittent and continuous infusion regimens in adults.

## 2. Materials and Methods

### 2.1. Study Design, Patients, and Data Collection

This study was designed as a retrospective, monocentric, and hospital-wide evaluation study in adults. The study was approved by the Ethics Committee Research UZ/KU Leuven (S65213).

All adult patients admitted to any ward at the University Hospitals Leuven between January 2019 and January 2021 and treated with either intermittent or continuous vancomycin infusion were eligible for inclusion. To cover a hospital-wide adult patient population, both patients admitted to the intensive care unit (ICU) wards and non-ICU wards were included. Patients were excluded if they did not provide a minimum of three vancomycin plasma concentrations measured consecutively during one treatment course. Patients with two consecutive concentrations measured after >72 h were also excluded.

Demographic, clinical, and laboratory data were collected from the patients’ medical files. For each patient, two randomly selected pairs of two consecutive vancomycin plasma concentrations were selected by a ‘Research Randomizer’ tool [16] as illustrated in Figure 1. For each pair of two consecutive plasma concentrations, the vancomycin dosing information was collected from the first vancomycin dose until the time of the second vancomycin concentration.

### 2.2. Bayesian Forecasting

A free web-based MIPD software, TDMx [17], was used to evaluate the model-based vancomycin concentrations and doses. Five different model-based approaches were evaluated. First, a single-model approach using the Goti model [18] was evaluated. The Goti model was developed using a large and heterogeneous set of hospitalized patients. This specific model was selected as it was demonstrated to be the most suitable single model to drive model-based vancomycin dosing in both hospitalized non-ICU patients [14] and ICU patients [13]. Next, two automated multi-model algorithms were also evaluated, i.e., the MSA and the MAA [15]. Both algorithms were recently developed and validated in a heterogeneous population and integrated into TDMx. In TDMx, MSA and MAA predictions are based on seven vancomycin PopPK models [18,19,20,21,22,23,24]. These seven PopPK models were developed in distinct populations covering a broad hospital setting, including hospitalized, critically ill, extremely obese, dialysis, sepsis, trauma, and post-cardiac surgery patients. Therefore, these vancomycin models were considered representative of the hospital-wide adult patient population investigated in our study. The characteristics of these vancomycin models were reported in detail previously [15]. All models included at least creatinine clearance or serum creatinine and total body weight as covariates. All of the covariates included in the seven vancomycin models were collected except for the Simplified Acute Physiology Score (SAPS) score, which is not collected at our institution.

Both multi-model approaches used all of the available models simultaneously to provide predictions. The prediction of the MSA was the prediction of the best-fitting model for an individual patient based on the goodness-of-fit of each model to the previous vancomycin concentration (i.e., how well did the model predict the previous vancomycin concentration). Similarly, the MAA considered the same previously collected data. However, instead of selecting the best-fitting model, the prediction of the MAA was based on the weighted predictions of all of the PopPK models. The MSA and MAA both required a criterion to quantify the individual model’s fit and to attribute the final weighting schemes.

Two different criteria to assess the models’ fit metrics were evaluated: the objective function value (OFV) and the squared prediction errors (SSE). These weighting schemes were described in detail previously [15]. The MSA and MAA were evaluated according to these two weighting schemes. As such, we evaluated five different model-based approaches in this study: the Goti model, the MSA with weighting according to the OFV (MSA_OFV_) or the SSE (MSA_SSE_), and the MAA with weighting according to the OFV (MAA_OFV_) or the SSE (MAA_SSE_).

### 2.3. Evaluation of the Model-Based Approaches

For each of the five above-mentioned model-based approaches, we evaluated the predictive performance regarding the vancomycin model-based predictions. Subsequently, the model-based approach with the best predictive performance was used to evaluate the model-based vancomycin doses compared with the standard TDM-based doses. Additionally, a sensitivity analysis was performed using the model-based approach with the worst predictive performance to evaluate the robustness of the model-predicted doses across the different model-based approaches.

Initially, the first vancomycin concentration of each pair was used to inform the model-based predictions as illustrated in Figure 1. The second vancomycin concentration was then used to evaluate the predictive performance of the model-based dosing by comparing the model-based predicted concentration with the observed vancomycin concentration. The model-predicted concentration was based on patient characteristics, previously administered doses (up to the second plasma concentration), and the previous and most recent vancomycin concentration.

The bias, relative bias (rBias), and relative root mean square error (rRMSE) were calculated to compare the individually predicted and observed second vancomycin concentration of each pair.
(1)$$\mathrm{Bias}=\frac{1}{n}\;\times \;\sum_{1}^{i}(predicted_{i\;}-observed_{i})$$
(2)$$\mathrm{rBias}=\frac{1}{n}\;\times \;\sum_{1}^{i}(\frac{predicted_{i}-observed_{i}}{observed_{i}})\;\times \;100\%$$
(3)$$\mathrm{rRMSE}=\sqrt{\frac{1}{n}\times \sum_{1}^{i}\;\frac{{\left(predicted_{i}-observed_{i}\right)}^{2}}{{\left(observed_{i}\right)}^{2}}\;}\;\times \;100\%$$
where *n* is the total number of second vancomycin concentrations.

The predictive performance was considered acceptable if the mean bias was ≤±2 mg/L for concentrations below 20 mg/L or if the rBias was ≤±10% for concentrations of 20 mg/L and higher, thereby meeting the analytical quality requirements of the Royal College of Pathologists of Australasia applied at our institution. The 95% confidence intervals of the mean (relative) bias also included 0. Additionally, an rRMSE as low as possible was aimed for.

At an individual level (i.e., for each pair separately), the acceptable bias was also evaluated according to the above-mentioned criteria. The performance was considered acceptable if the bias was ≤± 2 mg/L for concentrations below 20 mg/L or if the rBias was ≤±10% for concentrations of 20 mg/L and higher. The classification accuracy was also evaluated; it was defined as no change in exposure category (i.e., subtherapeutic, supratherapeutic, or therapeutic) between the predicted vancomycin concentration and the observed concentration. Therapeutic exposure was defined as vancomycin concentrations between 12.5–17.5 mg/L or 20–25 mg/L depending on intermittent or continuous infusion, respectively.

Next, the prescribed doses were compared with the model-based doses predicted to reach the target concentration of 15 mg/L and 22.5 mg/L according to the local hospital’s guidelines for intermittent and continuous infusion, respectively. In clinical practice, it is common to prescribe a dose that is considered to reach therapeutic exposure at steady state. In contrast, model-based dosing aims to attain therapeutic exposure as soon as possible, i.e., at the next dosing interval. Therefore, the model-based doses were predicted as the first two doses to be administered after the first concentration of each pair. The prescribed dose in clinical practice was compared with the model-predicted second dose—which was a steady-state dose—since the model-predicted first dose was often not representative of the steady-state dose. All model-predicted and prescribed doses were normalized to a twice-daily dosing regimen, i.e., doses were expressed as dose every 12 h (q12h).

The absolute differences were calculated and evaluated graphically by comparing observed versus individually predicted vancomycin concentrations. Additionally, the correlation between the model-predicted and prescribed doses was evaluated using the intraclass correlation coefficient (ICC), which is an index of agreement between two measurements [25]. The ICC values were reported according to the best practices for reporting ICC parameters [26]. The subgroup analyses were performed to investigate the dose differences according to the vancomycin exposure at the second concentration (i.e., subtherapeutic, supratherapeutic, or therapeutic exposure).

### 2.4. Statistics

Statistical and graphical analyses of the data were performed using IBM SPSS Statistics 27 and R (version 4.0.0, R Core Team, Vienna, Austria). The figures were constructed in Microsoft PowerPoint (version 2203, Microsoft 365 MSO, Washington, DC, USA) and R (version 4.0.0, R Core Team, Vienna, Austria). The data were reported as count and percentage or median and interquartile range (IQR) as appropriate. A Wilcoxon signed-rank test was performed to evaluate the differences between vancomycin concentrations in patients receiving intermittent or continuous infusions separately since different targets were aimed for in both groups. The statistical significance was defined using a two-sided *p*-value ≤ 0.05. In line with previous studies, we aimed to collect data from 150 patients to represent the adult hospital-wide population that received vancomycin at the University Hospitals Leuven. To further increase the sample size, two pairs of vancomycin concentrations were collected for each patient. As such, we considered data from 150 patients, corresponding to 300 pairs of two consecutive vancomycin concentrations, appropriate for performing a robust evaluation of the predictive performance and predicted doses of model-based vancomycin dosing.

## 3. Results

### 3.1. Clinical Data

In total, 616 vancomycin concentrations were collected from 154 patients. Each patient contributed four vancomycin concentrations (i.e., two pairs of two consecutive concentrations per patient). As shown in Table 1, both ICU and non-ICU patients receiving intermittent and continuous infusion therapy were included. The median [IQR] age was 63 [53; 72] years old. The reasons for hospital admission and vancomycin therapy were diverse, as illustrated in Table 1. On the day of the first vancomycin concentration, renal replacement therapy was present in 23 (7.5%) of all pairs.

Among the first vancomycin concentrations of each pair, 45.1% were within the therapeutic range, with 31.2% and 23.7% of the concentrations being subtherapeutic and supratherapeutic, respectively. Following the standard TDM-based dose optimization, 48.1% of the second vancomycin concentrations of each pair were in the therapeutic range. The proportion of the subtherapeutic concentrations decreased to 21.4%, in contrast with the proportion of the supratherapeutic concentrations, which increased to 30.5% (Table 2).

As shown in Table 2, the median vancomycin concentration was 15 mg/L and 21.3 mg/L in the first sample of each pair and increased significantly to 15.7 mg/L and 22.1 mg/L in the second sample in patients receiving intermittent (*p* = 0.004) and continuous infusions (*p* = 0.0003), respectively.

### 3.2. Evaluation of the Model-Based Approaches

#### 3.2.1. Predictive Performance

Across the five model-based approaches investigated in our study, the mean bias in the overall population ranged from −1.53 mg/L to −0.11 mg/L. Depending on the model-based approach, the mean rBias ranged from −5.36% to 3.18% in the overall population, and 95% confidence intervals were all within ±10% as illustrated in Figure 2. However, the 95% confidence intervals did not include 0 for the MSA_OFV_ approach. The rRMSE was ≤28.1% for all model-based approaches (Figure 2). The goodness-of-fit plots showed a clinically acceptable fit of the predicted concentrations with the observed concentrations across all five model-based approaches, as illustrated in Figure 3. At the individual level, acceptable rBias was found in 38–46.4% of all predicted concentrations (Appendix A). The classification accuracy was 47.7–55.5%, depending on the model-based approach.

The analyses in the subpopulations are shown in Appendix A. The performance remained similar or even better in non-ICU patients or patients receiving the intermittent vancomycin infusion. However, in ICU patients or patients receiving the continuous vancomycin infusion, the performance decreased slightly. In these patients, only the Goti model showed 95% confidence intervals of the mean rBias, including 0. Additionally, for two of the multi-model-based approaches (i.e., MSA_OFV_ and MAA_OFV_), the 95% confidence intervals exceeded ± 10%.

#### 3.2.2. Model-Predicted Vancomycin Doses Compared with Standard TDM-Based Doses

Based on the predictive performance, we selected the MAA_SSE_ approach as the model-based approach with the best predictive performance (defined as a combination of the lowest overall rBias and rRMSE). The dose differences were calculated with the doses predicted according to the MAA_SSE_ approach. The median differences between the second (i.e., steady-state) model-predicted dose q12h and the prescribed (i.e., standard TDM-based) dose q12h were 113 [−69; 427] mg, −70 [−208; 120] mg, and 40 [−84; 197] mg in the case of subtherapeutic, supratherapeutic, and therapeutic exposure at the time of the second vancomycin concentration, respectively. These differences are depicted in Figure 4, together with the differences between the first model-predicted dose and the prescribed dose. The differences with the first model-predicted dose were substantially larger than that of the second model-predicted dose.

Overall, the ICC showed a moderate correlation between the second model-predicted dose and the prescribed dose (0.656). The correlation remained moderate in the case of the subtherapeutic and therapeutic vancomycin concentrations, as illustrated by an ICC of 0.665 and 0.596, respectively. In the case of the supratherapeutic vancomycin concentrations, the ICC showed a good correlation (0.821).

For the sensitivity analysis, the MSA_OFV_ approach was selected as the approach with the worst performance as this was the only model-based approach that did not show an overall mean rBias, including 0 in its 95% confidence intervals. As shown in Figure 1, similar differences between the second model-predicted dose q12h and the prescribed dose q12h were observed with the MSA_OFV_ compared with the dose differences calculated with the MAA_SSE_ approach.

## 4. Discussion

In our study, we investigated TDM-based vancomycin dose optimization in a broad and diverse population of hospitalized adult patients receiving either intermittent or continuous vancomycin infusions. We demonstrated that a single-model approach or multi-model approach performs well, as illustrated by the acceptable predictive performance. Moreover, the model-based vancomycin dose predictions suggest a potential for increased therapeutic exposure compared with the standard TDM-based dosing.

A great deal of vancomycin PopPK models have been developed in both specific and general adult patient populations [13,14,27]. However, it remains challenging to select and implement the right model to guide the vancomycin dosing for an individual patient. Many models have failed to show good predictive performance outside the population they were developed in [13,14,27]. In our study, we showed that the Goti model, as well as different multi-model approaches, performs acceptably well in a broad set of hospitalized patients. The Goti model [18] as single-model approach is probably the simplest model-based approach and covers many of the standard hospitalized patients. Interestingly, we observed that the MAA_SSE_ approach showed slightly better performance. A potential explanation might be that the multi-model approach performs better in specific patient populations for which the Goti model is not suitable. It should be noted that we only used the most recent previous vancomycin concentration to inform the TDM-based dose optimization. Multiple previous concentrations were deemed unnecessary. This agreed with recent studies demonstrating that performance metrics were superior [27], or similar [14], when based on the most recent vancomycin concentration compared with a higher number of previous concentrations.

Recently, Heus and co-workers [27] found that the Okada [28] and Colin [29] PopPK models performed best in a set of non-ICU patients treated with continuous infusion vancomycin. They also found the multi-model approaches investigated in this study performed similarly well as these single models. Noteworthy, in their study, the Goti model showed overall high imprecision and overpredicted the observed vancomycin concentrations. Heus et al. related this discrepancy to the difference in the mode of administration and potential variation in assay methods for vancomycin and serum creatinine concentration. In the future, the performance of the multi-model approaches in the population investigated in our study might be further improved by including the Okada and Colin models. Still, in our study including both intermittent and continuous infusion, the Goti model performed well. Our results are in accordance with previous studies that found the Goti model to perform acceptably well with similar rBias and rRMSE in patients receiving intermittent or continuous infusions of vancomycin [13,14]. Interestingly, these studies also included dialysis patients, as in our study (23/308), whereas that of Heus and co-workers [27] excluded dialysis patients. We also performed subanalyses to identify potential subpopulations in which specific model-based approaches performed differently. Whereas the predictive performance remained similar in non-ICU patients or patients receiving an intermittent vancomycin infusion, the performance decreased slightly in ICU patients or patients receiving a continuous infusion. Nevertheless, the overall predictive performance was acceptable in the total population for four out of the five model-based approaches (Appendix A). In the future, the specific subpopulations in which the model-based approaches perform worse should be further expanded on. The benefit of including additional models in the multi-model approaches may also be investigated in order to make model-based vancomycin dosing even more widely applicable.

Although the acceptable performance at an individual level was only found in 38–46.4% of all predicted concentrations, the classification accuracy was higher (47.7–55.5%). It should be noted that we defined stricter criteria for acceptable performance than those of previous studies (i.e., rBias ± 10% versus rBias ± 20% [13,14,15] or 2.5 mg/L [27]). Using an rBias ± 20%, an acceptable bias was attained in 60.1–69.5% of all predicted concentrations, which overestimated the classification accuracy. Eventually, the MAA_SSE_ approach was selected to perform dose predictions, as overall it was the least biased and most precise approach. Moreover, the MAA_SSE_ approach showed good performance at an individual level and good classification accuracy compared with the other approaches.

Even though the vancomycin concentrations increased significantly after the standard TDM-based dose optimization, less than 50% of all second concentrations still showed therapeutic exposure (i.e., 48.1%). This agreed with a recent study revealing poor target attainment in adults with real-life standard TDM-based vancomycin dosing [10].

In our study, the model-predicted doses show only moderate to good correlation with prescribed doses. This finding illustrates a large opportunity for improvement in therapeutic exposure during vancomycin therapy that might be addressed with model-based TDM. The model-predicted doses suggest a potential increase in therapeutic exposure as suggested by the differences with the prescribed dose (Figure 4). In patients with subtherapeutic vancomycin exposure after the standard TDM-based dose optimization, the predicted doses are mostly larger than the prescribed doses. Hence, if the predicted doses had been administered to those patients, the subtherapeutic exposure may have been avoided in a substantial proportion of these patients. On the other hand, the lower doses were most frequently predicted in patients with supratherapeutic exposure after the standard TDM-based dose optimization. Therefore, less supratherapeutic exposure may also be expected from model-based dosing. Considering a dose modification of 125 mg or more as clinically relevant, the model-based vancomycin dosing would most frequently lead to a dose increase or a dose reduction in patients with a subtherapeutic or supratherapeutic second vancomycin concentration, respectively (Appendix A). In patients with a therapeutic second vancomycin concentration, the doses remained mostly unchanged. These findings were also confirmed in the sensitivity analysis when using the model-based approach with the worst predictive performance, i.e., the MSA_OFV_ approach. This finding strongly suggests that this is a benefit of the model-based vancomycin dosing compared with that of the standard TDM-based dosing.

It should be noted that the dose differences were calculated with the second model-predicted dose after the first TDM measurement. When using MIPD, the user should be aware that model-based predictions aim for direct target attainment, often resulting in relatively low or high doses immediately after a TDM measurement. In contrast, the subsequent (i.e., second) dose will be more reflective of the steady-state dose required to keep the exposure at the predefined target. This finding was confirmed by the large dose differences observed when comparing the first model-predicted dose with the prescribed dose (Figure 4). We opted to compare with the second model-based dose, as most often in clinical practice the clinicians will adapt the vancomycin dose to the new steady-state dose they believe is adequate for reaching the target exposure.

This study has several strengths. First, we intentionally performed our study in a broad and heterogeneous set of adult patients, hence increasing its external generalizability. Second, we evaluated both single and multi-model approaches. Third, we performed a comprehensive evaluation of the model-based approaches using goodness-of-fit plots, predictive performance, classification accuracy, and sensitivity analysis. Moreover, whereas previous studies only evaluated the predictive performance [13,14,15,27], we also evaluated model-predicted doses in relation to the exposure. Notwithstanding, several limitations to our study should also be taken into account. First, we need to acknowledge the limitations inherent to the retrospective and monocentric design of our study. Therefore, the findings may not necessarily be extrapolated to other centers, and prospective validation is needed. When prospectively validating model-based vancomycin dosing, one should be cautious of those patients for which predicted exposure is substantially over- or underestimated (i.e., inacceptable rBias at the individual level). These might represent patients with erroneous sampling or specific patients not appropriately covered by the included PopPK models. Second, we only included a subset of all available vancomycin PopPK models. We did not include the Okada [28] and Colin [29] models, which were recently found to perform the best in continuous-infusion vancomycin patients without dialysis [27]. Nevertheless, our analysis showed good overall predictive performance. Third, it should be noted that we did not include children, although children may also benefit from a similar model-based approach [10,29]. Fourth, our sample size of 154 patients is rather limited. However, our sample size is within the range of previous studies investigating the predictive performance of vancomycin model-based approaches. Interestingly, we performed an interim analysis—after inclusion of the first 65 patients—that showed results similar to the final results, hence supporting the robustness of our results [30]. Finally, we did not consider AUC/MIC targets as these are not commonly used at our institution and were therefore not available.

In conclusion, our study confirmed the predictive performance and illustrated the potential benefit of model-based vancomycin dosing compared with the standard TDM-based dosing in a broad and heterogeneous population. The Goti model, as well as most of the multi-model approaches, showed good overall performance. The MAA_SSE_ approach showed the best overall performance. Using this approach, the clinically relevant differences and moderate correlation between the model-based and prescribed dosing, along with the poor target attainment with the standard TDM-based dosing, suggested a large window of opportunity for the model-based TDM to increase therapeutic exposure. Implementation studies—preferably prospectively—of the model-based TDM in routine care are warranted.

## Figures and Tables

**Figure 1 pharmaceutics-14-01459-f001:**
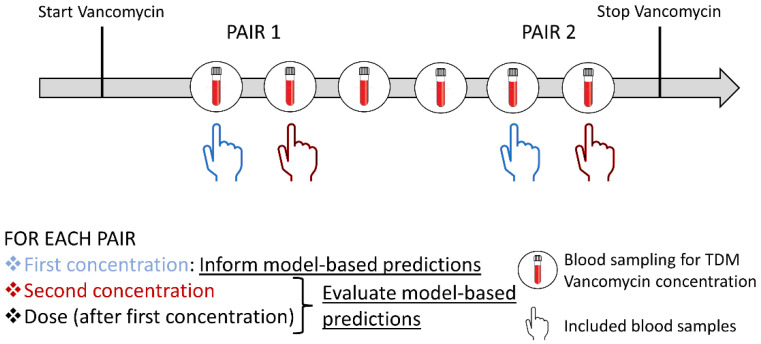
Schematic representation of the inclusion of two pairs of two consecutive vancomycin concentrations using a random example as illustration. The first concentration of each pair was used to inform the model-based predictions. The second concentration was blinded from the models and was used to evaluate the performance of the model-based prediction. The model-predicted dose after the first concentration was used to evaluate the model-based doses compared with the standard TDM-based doses.

**Figure 2 pharmaceutics-14-01459-f002:**
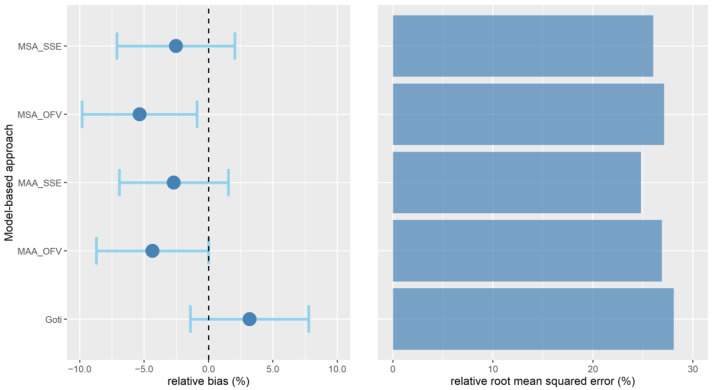
Overall relative bias and relative root mean square error of the predicted versus the observed second vancomycin concentration of each pair for the five model-based approaches investigated (i.e., the model selection algorithm [MSA] and the model averaging algorithm [MAA], according to the objective function value [OFV] and the squared prediction errors [SSE], and the single-model approach according to the Goti model). The blue dots represent the mean relative bias, and the blue error bars represent the 95% confidence intervals.

**Figure 3 pharmaceutics-14-01459-f003:**
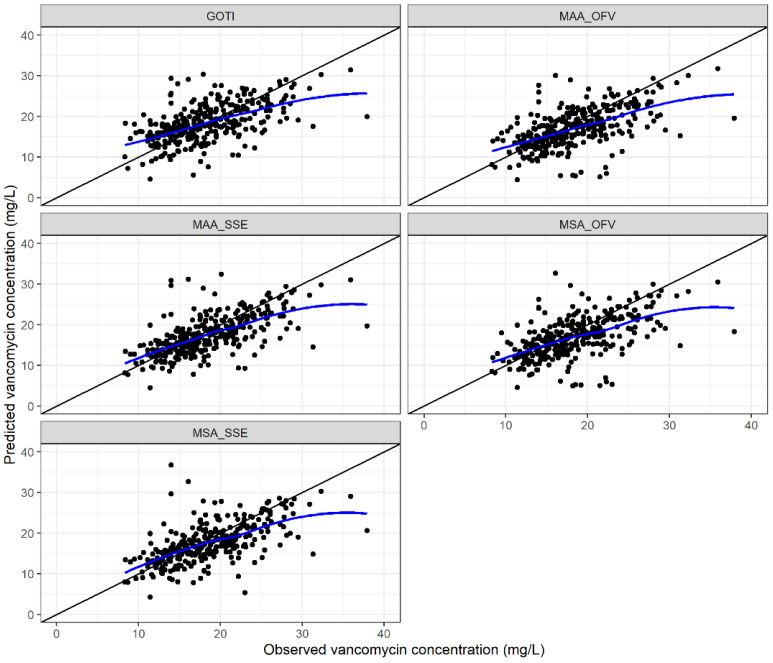
Goodness-of-fit plots of the predicted versus observed second vancomycin concentration of each pair for the five model-based approaches (i.e., the model selection algorithm [MSA] and the model averaging algorithm [MAA], according to the objective function value [OFV] and the squared prediction errors [SSE], and the single-model approach according to the Goti model). The blue line represents the local polynomial regression fit. The line of identity represents a perfect model fit.

**Figure 4 pharmaceutics-14-01459-f004:**
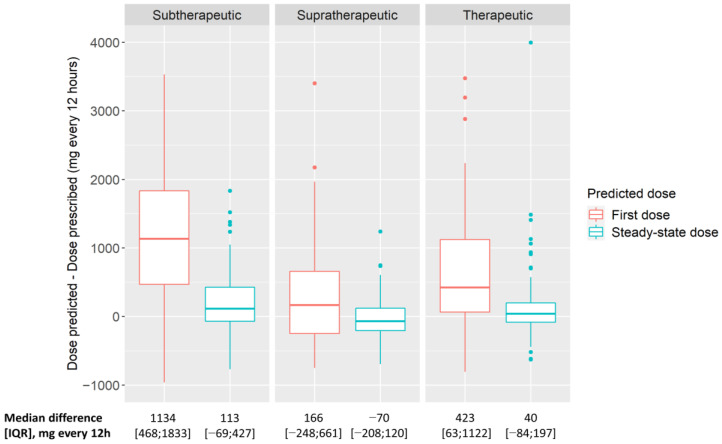
Boxplots of the difference between the vancomycin doses predicted by the MAA_SSE_ approach and the prescribed vancomycin doses, including the median (interquartile range) difference. The doses were normalized to a twice-daily dosing regimen, i.e., the doses were expressed as dose q12h. The differences were shown in three groups depending on the exposure at the time of the second vancomycin concentration of each pair (i.e., subtherapeutic, supratherapeutic, or therapeutic). The vancomycin doses were predicted based on the first vancomycin concentration of each pair of concentrations. The red and blue boxplots represent the dose differences based on the first and second dose predicted to reach therapeutic exposure, respectively. The therapeutic exposure was defined as concentrations between 12.5–17.5 mg/L or between 20–25 mg/L for intermittent or continuous infusion, respectively.

**Table 1 pharmaceutics-14-01459-t001:** Patient characteristics.

Per Patient			
	All (*n* = 154)	Intermittent (*n* = 95)	Continuous (*n* = 59)
Male, *n* (%)	103 (66.9)	68 (71.6)	35 (59.3)
Caucasian, *n* (%)/Afro-American, *n* (%)	149 (96.8)/5 (3.2)	90 (94.7)/5 (5.3)	59 (100)
Age (years), median [IQR]	63 [53; 72]	63 [55; 74]	60 [49; 68]
Weight (kg), median [IQR]	76 [64; 94]	77 [63; 95]	74 [65; 94]
Diabetes mellitus (type I, type II, and corticosteroid-induced), *n* (%)	43 (27.9)	32 (33.7)	11 (16.8)
Intensive care unit, *n* (%)	60 (39)	30 (31.6)	30 (50.8)
In-hospital mortality, *n* (%)	43 (27.9)	17 (17.3)	26 (44.1)
**Reason for Hospital Admission (** ** *n* ** **= 154)**			
Surgical, *n* (%)	54 (35.1)	45 (47.4)	9 (15.3)
Medical, *n* (%)	44 (28.6)	23 (24.2)	21 (35.6)
Emergency, *n* (%)	51 (33.1)	26 (27.4)	25 (42.4)
Others, *n* (%)	5 (3.2)	1 (1.1)	4 (6.8)
**Focus of the Infection (*n* = 154)**			
Respiratory, *n* (%)	21 (13.6)	10 (10.5)	11 (18.6)
Gastrointestinal, *n* (%)	7 (4.5)	5 (5.3)	2 (3.4)
Endocarditis, *n* (%)	6 (3.9)	3 (3.2)	3 (5.1)
Urinary, *n* (%)	3 (1.9)	3 (3.2)	0 (0)
Skin and soft tissue, *n* (%)	19 (12.3)	16 16.8)	3 (5.1)
Bone and joint, *n* (%)	24 (15.6)	20 (21.1)	4 (6.8)
Catheter-related, *n* (%)	18 (11.7)	9 (9.5)	9 (15.3)
Abdominal, *n* (%)	15 (9.7)	9 (9.5)	6 (10.2)
Postoperative, *n* (%)	10 (6.5)	7 (7.4)	3 (5.1)
Neutropenic fever, *n* (%)	25 (16.2)	8 (8.4)	17 (28.8)
Other, *n* (%)	6 (3.9)	5 (5.3)	1 (1.7)
**Clinical and Biochemical Data on the Day of the First Vancomycin Concentration Measurement**			
	**All** **(*n* = 308)**	**Intermittent** **(*n* = 190)**	**Continuous** **(*n* = 118)**
Serum creatinine (mg/dL), median [IQR]	0.83 [0.64; 1.22]	0.82 [0.61; 1.17]	0.84 [0.67; 1.34]
eGFR CKD-EPI (mL/min/1.73 m^2^), median [IQR]	87 [59; 104]	86 [61; 102]	88 [54.5; 107]
eCrCl CG (mL/min), median [IQR]	93 [58; 132]	90.2 [60; 132]	98 [51; 129]
Serum albumin (g/L) ^a^, median [IQR], *n*	31.2 [27.8; 34.6], 175	30.5 [26.8; 34.3], 62	31.9 [29; 34; 8], 113
Serum urea nitrogen (mg/dL), median [IQR]	31 [21; 55]	28 [20; 45]	39 [24; 77]
SOFA score ^b^, median [IQR], *n*	11 [7; 15], 120	8 [5; 12], 60	15 [11; 18], 60
Intermittent hemodialysis, *n* (%)	3 (1.0)	0 (0)	3 (2.5)
Intermittent peritoneal dialysis, *n* (%)	2 (0.6)	2 (1.1)	0 (0)
Continuous veno-venous hemofiltration, *n* (%)	18 (5.8)	5 (2.6)	13 (11)
Use of furosemide, *n* (%)	45 (14.6)	30 (15.8)	15 (12.7)

eCrCl CG: estimated creatinine clearance according to the Cockcroft–Gault equation; eGFR CKD-EPI: estimated glomerular filtration ratio according to the Chronic Kidney Disease Epidemiology Collaboration equation; IQR: interquartile range; n: count; SOFA: sequential organ failure assessment. ^a^ If available; ^b^ If ICU patient.

**Table 2 pharmaceutics-14-01459-t002:** Vancomycin concentrations (two pairs per patient) during intermittent (trough) and continuous infusion.

Vancomycin trough Concentrations during Intermittent Infusion (*n* = 190).	
First concentration (mg/L), median [IQR]	15.0 [12; 17.7]
Second concentration (mg/L), median [IQR]	15.7 [13.7; 18.3]
**Vancomycin Concentrations during Continuous Infusion (*n* = 118)**	
First concentration (mg/L), median [IQR]	21.3 [17.4; 23.5]
Second concentration (mg/L), median [IQR]	22.1 [19.3; 25;5]
**Exposure at Second Concentration (*n* = 308)**	
Therapeutic exposure ^a^, *n* (%)	148 (48.1)
Supratherapeutic exposure ^b^, *n* (%)	94 (30.5)
Subtherapeutic exposure ^c^, *n* (%)	66 (21.4)

^a^ 12.5–17.5 mg/L (intermittent) and 20–25 mg/L (continuous); ^b^ >17.5 mg/L (intermittent) and >25 mg/L (continuous); ^c^ <12.5 mg/L (intermittent) and <20 mg/L (continuous).

## Data Availability

The data are contained within the article and Appendix A. Additional data are available upon reasonable request.

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
