# Peer review of "Systematic Comparison of Hospital-Wide Standard and Model-Based Therapeutic Drug Monitoring of Vancomycin in Adults"

_pharmaceutics, 2022, doi:10.3390/pharmaceutics14071459_

Round 1

Reviewer 1 Report

In this manuscript the authors performed a rigorous evaluation of the predictive performance and predicted doses of a single-model and several multi-model approaches as compared to standard therapeutic drug monitoring-based vancomycin dosing, studying adult hospital patients treated with either intermittent or continuous vancomycin infusion. The study appears well designed, the results are well presented and appropriate conclusions are given. The manuscript is thorough and well written. I suggest that the manuscript is suitable for publication in Pharmaceutics with some minor changes to the text, as follows:   Line 38.  Define “TDM” Line 41.  Concentrations of what? Line 231.  Change “is” to “was” Line 232.  Change “is” to “was” Line 234.  Change “is” to “was” Line 545.  Italicise “Staphylococcus aureus” Lines 553-554.  Italicise “Staphylococcus aureus” Line 545.  Italicise “Staphylococcus aureus”    

Reviewer 2 Report

This study examines the optimization of vancomycin dosing by TDM in a broad and diverse group of hospitalized adult patients receiving intermittent or continuous infusion of vancomycin. Although considered lacking in novelty and clinical applicability, it exposes problems with the current model-informed TDM

About model-based prediction,…

1.      The characteristics and analysis methods of each model should be briefly summarized. I have no idea what factors affected the predictability of the current study.

2.      Are the patients used to establish the TDMx model consistent with the background of the current patients?

3.      Am I correct in assuming that TDMx has built-in parameters for continuous dosing?

4.      I don't understand the algorithm for MSA and MAA predictions - is it my understanding that one parameter prediction from 7 models and the blood concentration at a certain point in time is predicted? Has this prediction algorithm been validated (especially in patients with background factors comparable to our patient population)?

5.      We believe that the pharmacokinetic changes of vancomycin are not complex. Is each model predicted to account for renal function, fluid volume changes, extracellular fluid volume changes, and severity of infection (changes in inflammatory markers such as CRP)?

6.      There may be multiple causes of poor prediction. Are there any considerations regarding the factors that lead to poor predictions?

7.      Do the authors believe that there is a model that predicts well for all patients?

8.      When designing vancomycin dosing in clinical practice, do you recommend selecting a model that is tailored to the patient's background?

Lines 381-: The explanation of the methodology (about Goti model) should be stated in the methods section.

Lines 410-412: I believe that any discussion of the prediction of blood levels of vancomycin in hemodialysis patients is unnecessary in this study report.

Round 2

Reviewer 2 Report

No additional comments